# Chemical Variability and In Vitro Anti-Inflammatory Activity of Leaf Essential Oil from Ivorian *Isolona dewevrei* (De Wild. & T. Durand) Engl. & Diels

**DOI:** 10.3390/molecules26206228

**Published:** 2021-10-15

**Authors:** Didjour Albert Kambiré, Jean Brice Boti, Ahmont Claude Landry Kablan, Daouda Ballo, Mathieu Paoli, Virginie Brunini, Félix Tomi

**Affiliations:** 1UPR de Chimie Organique, Département de Mathématiques, Physique et Chimie, UFR des Sciences Biologiques, Université Péléforo Gon Coulibaly, Korhogo BP 1328, Côte d’Ivoire; dakambire@gmail.com (D.A.K.); kablanahmont@yahoo.fr (A.C.L.K.); 2Laboratoire de Constitution et Réaction de la Matière, UFR-SSMT, Université Félix Houphouët-Boigny, Abidjan BP 1328, Côte d’Ivoire; jeanbriceboti@hotmail.fr (J.B.B.); daoudaballo526@gmail.com (D.B.); 3Laboratoire Sciences Pour l’Environnement, Université de Corse—CNRS, UMR 6134 SPE, Route des Sanguinaires, 20000 Ajaccio, France; paoli_m@univ-corse.fr (M.P.); brunini_v@univ-corse.fr (V.B.)

**Keywords:** *Isolona dewevrei*, leaf essential oil, chemical variability, in vitro anti-inflammatory activity

## Abstract

The chemical variability and the in vitro anti-inflammatory activity of the leaf essential oil from Ivorian *Isolona dewevrei* were investigated for the first time. Forty-seven oil samples were analyzed using a combination of CC, GC(RI), GC-MS and ^13^C-NMR, thus leading to the identification of 113 constituents (90.8–98.9%). As the main components varied drastically from sample to sample, the 47 oil compositions were submitted to hierarchical cluster and principal components analyses. Three distinct groups, each divided into two subgroups, were evidenced. Subgroup I−A was dominated by (*Z*)-β-ocimene, β-eudesmol, germacrene D and (*E*)-β-ocimene, while (10βH)-1β,8β-oxido-cadina-4-ene, santalenone, *trans*-α-bergamotene and *trans*-β-bergamotene were the main compounds of Subgroup I−B. The prevalent constituents of Subgroup II−A were germacrene B, (*E*)-β-caryophyllene, (5αH,10βMe)-6,12-oxido-elema-1,3,6,11(12)-tetraene and γ-elemene. Subgroup II−B displayed germacrene B, germacrene D and (*Z*)-β-ocimene as the majority compounds. Germacrene D was the most abundant constituent of Group III, followed in Subgroup III−A by (*E*)-β-caryophyllene, (10βH)-1β,8β-oxido-cadina-4-ene, germacrene D-8-one, and then in Subgroup III−B by (*Z*)-β-ocimene and (*E*)-β-ocimene. The observed qualitative and quantitative chemical variability was probably due to combined factors, mostly phenology and season, then harvest site to a lesser extent. The lipoxygenase inhibition by a leaf oil sample was also evaluated. The oil IC_50_ (0.020 ± 0.005 mg/mL) was slightly higher than the non-competitive lipoxygenase inhibitor NDGA IC_50_ (0.013 ± 0.003 mg/mL), suggesting a significant in vitro anti-inflammatory potential.

## 1. Introduction

*Isolona* Engl. is a genus belonging to the Annonaceae family and comprising about 20 species, widely distributed in tropical rain forests of West and Central Africa, and Madagascar. Five species of this genus grow wild in Côte d’Ivoire: *I. cooperi*, *I. campanulata*, *I. soubreana*, *I. deightonii* and *I. dewevrei* [1,2].

*Isolona dewevrei* (De Wild. & T. Durand) Engl. & Diels (synonym: *Monodora dewevrei* De Wild. & T. Durand) is an evergreen shrub or a tree that can reach 15 m in height. It has narrowly obovate to obovate or elliptic to narrowly elliptic leaves that are 10–17 cm long and 4–7 cm wide, with an acuminated apex. The inflorescences appear on leafy branches and sometimes on older ones, whereas the fruits are ovoid (6–7 cm long, 4–5 cm in diameter), smooth but very finely ribbed, glabrous, green and yellow at maturity [1]. Although no ethno-pharmacological use of *I. dewevrei* was reported in the literature, other *Isolona* species (*I. cooperi* and *I. campanulata*) are traditionally used as herbal medicine in Côte d’Ivoire to treat bronchial ailments, skin diseases, hematuria and infertility, and to facilitate childbirth [1,2].

Previous studies of some solvent extracts reported several alkaloids, sterols and sesquiterpenes, as well from *I. maitlandii*, *I. cauliflora*, *I. pilosa*, *I. campanulata, I. zenkeri* and *I. hexaloba*, as from *I. cooperi*, from Côte d’Ivoire [3,4,5,6,7,8,9,10,11]. The chemical composition of the leaf, stem bark, root bark essential oils from *I. cooperi* and leaf essential oil from *I. campanulata* were also determined. The leaf and stem bark oils from *I. cooperi* were dominated by (*Z*)-β-ocimene and γ-terpinene, while the prevalent compounds of the root bark were 5-isopentenylindole and (*E*)-β-caryophyllene [12]. The leaf essential oil from *I. campanulata* was a sesquiterpene-rich oil. Its main compounds were either eudesm-5-en-11-ol, or (*E*)-β-caryophyllene and α-humulene [13].

In our previous works, we investigated and reported for the first time the chemical composition of the leaf, root and stem bark essential oils from *I. dewevrei*. The structure of germacrene D-8-one, a new natural compound, was elucidated after isolation from the stem bark essential oil. The major compounds of the root oil were cyperene and camphene, followed by 5-isopentenylindole, β-elemene and (*Z*)-α-bisabolene, whereas the stem bark oil was dominated by cyperene associated with β-elemene, (*Z*)-α-bisabolene, (*E*)-β-caryophyllene and α-copaene [14]. The leaf oil studies led to the isolation of seven new sesquiterpenes, characterized as 6,12-oxido-germacra-1(10),4,6,11(12)-tetraene (5αH,10βMe)-6,12-oxido-elema-1,3,6,11(12)-tetraene, germacra-1(10),4,7(11)-trien-6,12-γ-lactone, (1βH,5βH)-6,12-oxido-guaia-6,10(14),11(12)-trien-4α-ol, (10βH)-1β,8β-oxido-cadina-4-ene, (7βH)-germacrene D-8α-ol and (7βH)-germacrene D-8β-ol. NMR data of cadina-1(10),4-dien-8β-ol were also reported for the first time [15,16]. The studied leaf oil samples evidenced three chemical compositions dominated either by germacrene B, (5αH,10βMe)-6,12-oxido-elema-1,3,6,11(12)-tetraene, germacrene D, (*Z*)-β-ocimene, γ-elemene and (*E*)-β-caryophyllene, or by germacrene D, germacrene D-8-one, (10βH)-1β,8β-oxido-cadina-4-ene, (7βH)-germacrene D-8α-ol and cadina-1(10),4-dien-8β-ol, or by (10βH)-1β,8β-oxido-cadina-4-ene, (*E*)-β-caryophyllene, germacrene D, cadina-1(10),4-dien-8β-ol and caryophyllene oxide. Therefore, continuing our study on the chemical characterization of essential oils from Côte d’Ivoire [17,18,19,20,21], the chemical variability of the leaf essential oil of *I. dewevrei* was evaluated by investigating a larger number of oil samples (47). The in vitro anti-inflammatory activity of this essential oil was tested too.

## 2. Results and Discussion

The chemical composition of 47 leaf essential oil samples from *I. dewevrei* growing wild in Côte d’Ivoire was investigated in order to highlight possible variability. Oil samples were isolated by the hydrodistillation of fresh leaves collected at six sites of the Bossématié forest (Eastern Côte d’Ivoire) and two sites of the Haut-Sassandra forest (Western Côte d’Ivoire). The extraction yields calculated on a weight basis (*w/w*) were in the range of 0.094–0.506%. Analyses were carried out using a combination of GC(RI), GC-MS and ^13^C-NMR, following a computerized method developed at the University of Corsica [15,22]. Most of the constituents were identified by the three techniques, including the identification by ^13^C-NMR of components present at a content as low as 0.4–0.5% and compiled in our laboratory-made ^13^C-NMR spectral data library. However, several minor compounds remained unidentified despite the use of these complementary techniques. Hence, samples S2 (4.115 g), S14 (4.820 g), S24 (3.702 g), S44 (4.226 g) and S47 (3.910 g), which displayed qualitative and quantitative variations of the main constituents and the minor unidentified compounds, were separately submitted to a detailed analysis by column chromatography (CC). The CC fractions were then analyzed by the three above techniques. Finally, the 47 oil compositions were subjected to a statistical analysis, and the in vitro anti-inflammatory activity of the single monoterpene-rich oil sample (S44) was evaluated.

### 2.1. Detailed Analysis of Essential Oil Samples S2, S14, S24, S44 and S47 

The leaf oil samples S2, S14, S24, S44 and S47 were representative of the whole sampling and displayed different chromatographic profiles with several unidentified minor components. They were separately fractionated by silica gel column chromatography, and seven fractions were eluted for each sample using a gradient of solvents, *n*-pentane/diethyl ether. Fractions F1 and F2 contained non-polar compounds, while medium polar components were in fractions F3–F6 and polar constituents in fraction F7. The CC fractions analysis by GC(RI), GC-MS and ^13^C-NMR led to the identification of various minor constituents along with the majority compounds of the samples 

Special attention was paid to the identification of some compounds. For instance, eight sesquiterpenes previously isolated and characterized from the leaf essential oil of *I. dewevrei* were identified by their reported ^13^C-NMR data and retention indices [15,16]: (5αH,10βMe)-6,12-oxido-elema-1,3,6,11(12)-tetraene (fractions F3 from samples S14 and S47; 48.5 and 39.2%, respectively), 6,12-oxido-germacra-1(10),4,6,11(12)-tetraene (fraction F3 from sample S14; 7.2%), (1βH,5βH)-6,12-oxido-guaia-6,10(14),11(12)-trien-4α-ol (fraction F5 from sample S14; 56.4%), germacra-1(10),4,7(11)-trien-6,12-γ-lactone (fraction F6 from sample S14; 19.3%), (10βH)-1β,8β-oxido-cadina-4-ene (fraction F4 from samples S2, S24 and S44; 34.7, 18.6 and 14.7%, respectively), (7βH)-germacrene D-8α-ol (fraction F5 from sample S24; 29.4%), (7βH)-germacrene D-8β-ol (fraction F5 from sample S24; 9.8%) and cadina-1(10),4-dien-8β-ol (fraction F5 from sample S24; 39.2%). Similarly, germacrene D-8-one previously isolated from the stem bark essential oil of the same species was identified in fraction F3 from sample S24 (28.6%) [14].

Compounds bearing the elemane and germacrane skeletons and those co-eluting during the GC analysis needed special attention too. Indeed, under GC and GC-MS thermal conditions, germacrene compounds bearing the cyclodeca-1,5-diene sub-structure partially or totally rearranged to the corresponding elemanes (Cope transposition) [23,24]. This was the case for 6,12-oxido-germacra-1(10),4,6,11(12)-tetraene and (5αH,10βMe)-6,12-oxido-elema-1,3,6,11(12)-tetraene, and then germacrene B and γ-elemene. Therefore, their correct contents were obtained by a combination of GC(FID) and ^13^C-NMR as previously reported [15,16]. However, δ-elemene and β-elemene were really secondary metabolites produced by the plant and not rearranged products since germacrene A and germacrene C were not detected by ^13^C-NMR in the samples. The diastereoisomers (7βH)-germacrene D-8α-ol and (7βH)-germacrene D-8β-ol co-eluted on apolar and polar columns during the GC analysis. Thus, the ^13^C-NMR analysis allowed their identification, and their relative contents were assessed by a combination of GC(FID) and ^13^C-NMR [15,16].

The detailed analysis of essential oil samples S2, S14, S24, S44 and S47, by a combination of chromatographic and spectroscopic techniques led to the identification of a total of 99 compounds representing, respectively, 97.8, 96.4, 93.8, 98.9 and 98.1% of their whole composition (Table 1). Oil samples S2, S14, S24 and S47 were dominated by sesquiterpenes (76.0–92.5%), whereas sample S44 was monoterpene rich (71.2%). Significant variations appeared relative to the majority compounds. Indeed, the main constituents of oil sample S2 were santalenone (13.0%), *trans*-α-bergamotene (11.7%), *trans*-β-bergamotene (10.9%), (10βH)-1β,8β-oxido-cadina-4-ene (9.3%), β-bisabolene (8.7%) and α-santalene (8.5%). Sample S14 was dominated by germacrene B (20.8%), followed by (5αH,10βMe)-6,12-oxido-elema-1,3,6,11(12)-tetraene (10.3%), (*Z*)-β-ocimene (7.8%), germacrene D (7.5%), γ-elemene (6.9%) and (*E*)-β-caryophyllene (6.8%). Germacrene D (23.6%), germacrene D-8-one (8.7%), (7βH)-germacrene D-8-α-ol (7.8%), cadina-1(10),4-dien-8β-ol (7.6%) and (10βH)-1β,8β-oxido-cadina-4-ene (7.3%) were the most prevalent constituents of sample S24. Sample S44 was largely dominated by (*Z*)-β-ocimene (36.6%) and (*E*)-β-ocimene (30.5%), followed by germacrene D (13.8%), while β-eudesmol (22.5%), germacrene D (16.3%), α-eudesmol (12.4%), (*E*)-β-caryophyllene (12.4%) and (*Z*)-β-ocimene (10.2%) were the main compounds of sample S47. Obviously, the chemical compositions of the five essential oil samples displayed qualitative and quantitative variations (Table 1).

### 2.2. Chemical Variability of Leaf Essential Oil from I. dewevrei

The chemical variability of the leaf essential oil from *I. dewevrei* was evaluated throughout 47 samples collected from shrubs at different phenological stages, at six sites of the Bossématié forest (Eastern Côte d’Ivoire) and two sites of the Haut-Sassandra forest (Western Côte d’Ivoire), during the rainy and dry seasons. In total, 113 constituents accounting for 90.8–98.9% of the samples’ whole compositions were identified (Appendix A). All the samples except one (46/47) were largely dominated by sesquiterpenes (57.3–92.5%). Monoterpenes were prevalent in a single sample (71.2%), and the main components varied drastically from sample to sample: (*Z*)-β-ocimene (0.1–36.6%), germacrene D (0.2–34.0%), (*E*)-β-ocimene (0–30.5%), germacrene B (0.1–24.8%), β-eudesmol (0–22.5%), (10βH)-1β,8β-oxido-cadina-4-ene (0–13.7%), (5αH,10βMe)-6,12-oxido-elema-1,3,6,11(12)-tetraene (0–13.7%), limonene (tr–13.0%), santalenone (0–13.0%), (*E*)-β-caryophyllene (1.7–12.4%), α-eudesmol (0–12.4%), *trans*-α-bergamotene (0–11.7%), caryophyllene oxide (0–11.5%) and *trans*-β-bergamotene (0–10.9%). Therefore, the 47 essential oil compositions were subjected to a hierarchical cluster analysis (HCA) and principal component analysis (PCA) to highlight possible chemical variability.

The dendrogram from the HCA revealed three distinct groups within the 47 investigated oil samples: Group I (9 samples), Group II (19 samples) and Group III (19 samples), each consisting of two subgroups (Figure 1). The first principal factor of the PCA (F1: 44.31%), the second (F2: 22.52%) and the third (F3: 13.01%) accounted for 79.12% of the total variance of the chemical composition. The PCA map of the samples’ distribution relative to the principal axes F1 and F3 (57.32%) confirmed the three chemical composition groups (Figure 2). The Groups II and III were more homogenous than Group I. The mean contents (M) and the standard deviation (SD) of the majority compounds of the different subgroups are reported in Table 2.

The Subgroup I−A (four oil samples) was dominated by (*Z*)-β-ocimene (M = 18.0%, SD = 12.5%), β-eudesmol (M = 13.4%, SD = 9.7%), germacrene D (M = 12.6%, SD = 3.0%), (*E*)-β-ocimene (M = 11.4%, SD = 12.8%), (*E*)-β-caryophyllene (M = 8.0%, SD = 2.5%) and α-eudesmol (M = 7.6%, SD = 5.5%). The major compounds of Subgroup I−B (five oil samples) were (10βH)-1β,8β-oxido-cadina-4-ene (M = 10.9%, SD = 2.1%), santalenone (M = 9.0%, SD = 2.7%), *trans*-α-bergamotene (M = 7.8%, SD = 2.5%), *trans*-β-bergamotene (M = 7.8%, SD = 2.0%), α-santalene (M = 5.2%, SD = 1.8%) and β-bisabolene (M = 5.0%, SD = 2.6%). Santalenone, *trans*-α-bergamotene, *trans*-β-bergamotene, α-santalene and β-bisabolene, present in Subgroup I−B, were absent from Subgroup I−A and did not exceed 1.1% in Groups II and III. β-eudesmol and α-eudesmol did not exceed 0.2% in Group II and were absent from Subgroup III−A.

The Group II is characterized by its high content of germacrene B, (5αH,10βMe)-6,12-oxido-elema-1,3,6,11(12)-tetraene, γ-elemene and limonene. The main constituents of Subgroup II−A, which consisted of 13 oil samples, were germacrene B (M = 18.5%, SD = 4.1%), (*E*)-β-caryophyllene (M = 9.0%, SD = 1.8%), (5αH,10βMe)-6,12-oxido-elema-1,3,6,11(12)-tetraene (M = 7.8%, SD = 3.1%), γ-elemene (M = 6.3%, SD = 1.3%), (*Z*)-β-ocimene (M = 5.9%, SD = 2.7%), germacrene D (M = 5.7%, SD = 2.0%) and limonene (M = 4.4%, SD = 3.4%). The Subgroup II−B (six oil samples) was also dominated by germacrene B (M = 17.2%, SD = 4.1%), followed by germacrene D (M = 13.6%, SD = 5.2%), (*Z*)-β-ocimene (M = 13.2%, SD = 5.2%), (*E*)-β-caryophyllene (M = 7.5%, SD = 2.5%), (5αH,10βMe)-6,12-oxido-elema-1,3,6,11(12)-tetraene (M = 6.3%, SD = 4.4%), (*E*)-β-ocimene (M = 4.6%, SD = 1.6%) and γ-elemene (M = 4.2%, SD = 2.4%).

Group III differed from Groups I and II by its important proportions of germacrene D-8-one, (7βH)-germacrene D-8α-ol and cadina-1(10),4-dien-8β-ol. The Subgroup III−A (11 oil samples) displayed germacrene D (M = 24.1%, SD = 3.4%), (*E*)-β-caryophyllene (M = 8.2%, SD = 2.8%), (10βH)-1β,8β-oxido-cadina-4-ene (M = 6.4%, SD = 0.5%), germacrene D-8-one (M = 6.1%, SD = 2.2%), cadina-1(10),4-dien-8β-ol (M = 5.7%, SD = 1.2%), (7βH)-germacrene D-8-α-ol (M = 4.7%, SD = 2.3%) and (*Z*)-β-ocimene (M = 4.5%, SD = 1.6%) as the most prevalent compounds. Finally, germacrene D (M = 27.6%, SD = 4.1%), followed this time by (*Z*)-β-ocimene (M = 12.1%, SD = 2.0%), (*E*)-β-ocimene (M = 7.0%, SD = 1.6%), (*E*)-β-caryophyllene (M = 5.2%, SD = 2.1%) and (10βH)-1β,8β-oxido-cadina-4-ene (M = 4.0%, SD = 2.8%), were the majority components of Subgroup III−B (eight oil samples).

This study demonstrated both qualitative and quantitative variations of constituents within the detected groups and subgroups. The observed chemical variability of the leaf essential oil from *I. dewevrei* was probably due to several external factors, although genetic factors could not be completely excluded. As all samples were collected from shrubs at different phenological stages, at six sites of the Bossématié forest (Sites 1 to 6; Eastern Côte d’Ivoire) and two sites of the Haut-Sassandra forest (Sites 7 and 8; Western Côte d’Ivoire), during the rainy and dry seasons, the effect of phenology, season and harvest site on samples’ distribution into groups was examined (Figure 3). Regarding the phenology, samples collected from shrubs bearing flowers are all included in Group I, where they constituted the Subgroup I−A. Moreover, Group II is exclusively constituted of samples collected from shrubs bearing neither flowers nor fruits, while all the samples collected from shrubs bearing fruits are included in Group III. Therefore, the phenology appeared to be an important factor explaining the chemical variability of the leaf essential oil of the plant. From a seasonal point of view, samples collected during the dry season are more differenciated than those harvested during the rainy season. Indeed, the three barycenters of the samples collected during the dry season, corresponding to the three chemical groups, are furthest from the origin. In addition, Subgroups I−B, II−B and III−B exclusively consisted of samples collected during the rainy season, whereas samples from Subgroup I−A were harvested during the dry season. The effect of seasons on the chemical variability of *I. dewevrei* leaf oil could not therefore be overlooked. Although all the harvest sites are located in the same climatic zone (mesophilic sector; *Celtis* spp., *Triplochiton scleroxylon* and its variant of *Nesogordonia papaverifera* and *Khaya ivorensis* forests) with the same soil type (ferrallitic), the sites’ effect on the chemical variability was noticeable. In fact, the Subgroup I−A consisted of samples from Site 8, while those collected at Sites 3 and 4 were all included in Subgroup II−A. Samples from Site 7 are all included in Subgroup III−B, whereas those from Site 2 belonged to Group III. The harvest sites being sufficiently distant, their effect on the chemical variability could be related to possible microclimates or genetic differences. It could finally be argued that phenology, season and harvest site really impacted the chemical variability of the leaf essential oil from *I. dewevrei*.

### 2.3. Comparision to Literature Data

The chemical composition of the leaf essential oil from *I. dewevrei* differed considerably from that from *I. cooperi* and *I. campanulata*. Indeed, unlike leaves of *I. dewevrei* which produced a sesquiterpene-rich oil, the leaf oil from *I. cooperi* was largely dominated by monoterpenes: (*Z*)-β-ocimene (16.9–71.1%), γ-terpinene (0.6–26.3%), α-terpinene (0.3–12.7%), δ-3-carene (up to 8.8%), 5[(*Z*)-hexylidene]-5*H*-furan-2-one (1.2–7.8%) and massoia lactone (0.9–4.2%) [12]. Only (*Z*)-β-ocimene (0.1–26.7%) was a major compound of *I. dewevrei* samples, while γ-terpinene, α-terpinene and δ-3-carene did not exceed 1.1%. Moreover, 5[(*Z*)-hexylidene]-5*H*-furan-2-one and massoia lactone were not detected in *I. dewevrei* samples. The leaf oil from *I. campanulata*, on the other hand, was dominated by (*E*)-β-caryophyllene (8.2–30.5%), eudesm-5-en-11-ol (0.1–20.0%) and α-humulene (3.8–17.5%) [13]. Although (*E*)-β-caryophyllene (1.7–12.4%) was predominant in some *I. dewevrei* samples, its contents remained lower. In addition, eudesm-5-en-11-ol was completely absent from the leaf oil of *I. dewevrei*, while α-humulene was present in relatively small proportions (0.9–2.9%). Most of the other major constituents of *I. dewevrei* leaf oil were either absent or present at low contents in *I. cooperi* and *I. campanulata* leaf essential oils.

### 2.4. Evaluation of In Vitro Anti-Inflammatory Activity

The in vitro anti-inflammatory potential of *I. dewevrei* leaf essential oil (S44) was evaluated by determining its ability to inhibit lipoxygenases (LOX). Indeed, LOXs are nonheme iron-containing dioxygenases that convert linoleic, arachidonic and other polyunsaturated fatty acid into biologically active metabolites involved in the inflammatory and immune responses. Several inflammatory processes such as arthritis, bronchial asthma and cancer are associated with an important production of leukotrienes catalyzed by the LOX pathway from arachidonic acid [31,32,33,34]. The inhibition of the LOX pathway with inhibitors of LOX would prevent the production of leukotrienes and therefore could constitute a therapeutic target for the treating of human inflammation-related diseases. Thus, the search for new LOX inhibitors appears as critical because many exhibit significant in vitro anti-inflammatory activity.

The ability of *I. dewevrei* leaf essential oil to inhibit soybean lipoxygenase was determined as an indication of potential in vitro anti-inflammatory activity. *I. dewevrei* leaf essential oil exhibited an inhibition of LOX activity (Table 3). The percentage of inhibition increases with the concentration of the oil i.e., 10.3% at 0.005 mg/mL to 51.5% at 0.020 mg/mL of the essential oil. The IC_50_ values (concentration at which 50% of the lipoxygenase was inhibited) were determined for the *I. dewevrei* leaf essential oil and for the non-competitive inhibitor of lipoxygenase, the nordihydroguaiaretic acid (NDGA) (Table 3), usually used as a reference in in vitro anti-inflammatory assays [32,33,34]. Data showed that the IC_50_ value of *I. dewevrei* leaf essential oil (0.020 ± 0.005 mg/mL) is slightly higher than the IC_50_ value of NDGA (0.013 ± 0.003 mg/mL). The low ratio between the two values of IC_50_ (*I. dewevrei* leaf essential oil vs. NDGA) makes it possible to consider this essential oil as a high inhibitor of the LOX activity, suggesting an in vitro anti-inflammatory potential [35].

## 3. Material and Methods

### 3.1. Plant Material

The fresh leaves’ samples were collected on individual *I. dewevrei* shrubs at different phenological stages, at six sufficiently distant sites (Sites 1 to 6) of the Bossématié forest, Region of Abengourou, Eastern Côte d’Ivoire, and at two sites (Sites 7 and 8) of the Haut-Sassandra forest, Western Côte d’Ivoire. Geographical coordinates: Site 1 (6°31′34.7″ N and 3°28′15.0″ W), Site 2 (6°29′26.0″ N and 3°29′11.7″ W), Site 3 (6°27′44.9″ N and 3°32′25.5″ W), Site 4 (6°25′44.5″ N and 3°32′40.7″ W), Site 5 (6°26′07.2″ N and 3°28′27.2″ W), Site 6 (6°23′43.4″ N and 3°25′59.4″ W), Site 7 (6°53′40.2″ N and 6°55′36.3″ W) and Site 8 (6°54′52.7″ N and 6°57′21.1″ W). The harvest took place on one hand during the dry season (March 2016, January 2017 and February 2021) and during the rainy season on the other hand (April 2016, August 2016 and July 2020) (Appendix A). Plant material was authenticated by botanists from the Centre Suisse de Recherches Scientifiques (CSRS) and Centre National de Floristique (CNF) Abidjan, Côte d’Ivoire. A voucher specimen was deposited at the herbarium of CNF, Abidjan, with the reference LAA 12874.

### 3.2. Essential Oil Isolation and Fractionation

The essential oil samples were obtained by the hydrodistillation of fresh leaves for 3 h each, using a Clevenger-type apparatus. Yields were calculated from fresh material (*w/w*). Plant material and essential oil extraction data are reported in Appendix A. The leaf essential oil samples S2 (4.115 g), S14 (4.820 g), S24 (3.702 g), S44 (4.226 g) and S47 (3.910 g) were separately chromatographed on a column with silica gel (Acros Organics, Waltham, MA, USA, 60–200 μm, 120 g each, except S14, 150 g), using a gradient of solvents, distilled *n*-pentane (VWR Chemicals, Radnor, PA, USA, 99%)/diethyl ether (VWR Chemicals, 100.0%) of increasing polarity (P/DE, 100/0 to 0/100). Seven fractions were eluted for each oil sample: F1 and F2 (eluted with *n*-pentane) contained hydrocarbons; F3–F6 (eluted with P/DE mixtures) contained medium polar compounds; F7 (eluted with diethyl ether) contained polar compounds. The respective weights of the fractions from the different samples are reported in the Table 4.

### 3.3. Gas Chromatography

Analyses were performed on a Clarus 500 PerkinElmer Chromatograph (PerkinElmer, Courtaboeuf, France), equipped with a flame ionization detector (FID) and two fused-silica capillary columns (50 m × 0.22 mm, film thickness 0.25 µm), BP-1 (polydimethylsiloxane) and BP-20 (polyethylene glycol). The oven temperature was programmed from 60 °C to 220 °C at 2 °C/min and then held isothermal at 220 °C for 20 min; injector temperature: 250 °C; detector temperature: 250 °C; carrier gas: hydrogen (0.8 mL/min); split: 1/60; injected volume: 0.5 µL. Retention indices (RI) were determined relative to the retention times of a series of n-alkanes (C8–C29) with a linear interpolation (« Target Compounds » software from PerkinElmer). The relative response factor (RFF) of each compound was calculated according to the International Organization of the Flavor Industry (IOFI)-recommended practice for the use of predicted relative response factors for the rapid quantification of volatile flavoring compounds by GC(FID) [36]. Methyl octanoate was used as an internal reference, and the relative proportion of each constituent (expressed in g/100 g) was calculated using the weight of the essential oil and reference, peak area and relative response factors (RRF).

### 3.4. Gas Chromatography—Mass Spectrometry in Electron Impact Mode

The essential oil samples and all fractions of chromatography were analyzed with a Clarus SQ8S PerkinElmer TurboMass detector (quadrupole), directly coupled with a Clarus 580 PerkinElmer Autosystem XL (PerkinElmer, Courtaboeuf, France), equipped with a Rtx-1 (polydimethylsiloxane) fused-silica capillary column (60 m × 0.22 mm i.d., film thickness 0.25 µm). The oven temperature was programmed from 60 to 230 °C at 2°/min and then held isothermal for 45 min; injector temperature, 250 °C; ion-source temperature, 250 °C; carrier gas, He (1 mL/min); split ratio, 1:80; injection volume, 0.2 µL; ionization energy, 70 eV. The electron ionization (EI) mass spectra were acquired over the mass range 35–350 Da.

### 3.5. Nuclear Magnetic Resonance

All ^13^C-NMR spectra were recorded on a Bruker AVANCE 400 Fourier transform spectrometer (Bruker, Wissembourg, France) operating at 100.623 MHz for ^13^C, equipped with a 5 mm probe, in CDCl_3_, with all shifts referred to via an internal TMS. The following parameters were used: pulse width = 4 µs (flip angle 45°); relaxation delay D1 = 0.1 s, acquisition time = 2.7 s for a 128 K data table with a spectral width of 25,000 Hz (250 ppm); CPD mode decoupling; digital resolution = 0.183 Hz/pt. The number of accumulated scans was 3000 for each sample or fraction (40 mg, when available, in 0.5 mL of CDCl_3_).

### 3.6. Identification of Individual Components

Identification of the individual components was carried out: (i) by a comparison of their GC retention indices on apolar and polar columns, with those of reference compounds [25,37]; (ii) by computer matching against commercial mass spectral libraries [37,38,39]; (iii) by a comparison of the signals in the ^13^C-NMR spectra of the samples and fractions with those of reference spectra compiled in the laboratory spectral library, with the help of a laboratory-made software [15,22]. This method allowed for the identification of individual components of the essential oil at contents as low as 0.4–0.5%.

### 3.7. Statistical Analysis

The chemical compositions of 47 leaf essential oil samples from *I. dewevrei* were submitted to a hierarchical cluster analysis (HCA) and principal component analysis (PCA) using XLSTAT software (Addinsoft, Paris, France) [40]. Only constituents in a concentration higher than 1.0% were used as variables for the PCA analysis. The aptitude of the complete correlation matrix was checked by the Kaiser–Meyer–Olkin criterion. The HCA and dendrogram were made with dissimilarity matrices calculated in Euclidean distance, and the average link was the aggregation method systematically chosen.

### 3.8. In Vitro Anti-Inflammatory Capacity of Isolona dewevrei Leaf Essential Oil

The in vitro anti-inflammatory capacity of *I. dewevrei* leaf essential oil was evaluated by an in vitro lipoxygenase inhibition assay [41,42,43]. The in vitro analysis for lipoxygenase inhibitory activity was performed using Lipoxidase type I-B (Lipoxygenase, LOX, EC 1.13.11.12) from Glycine max (soybean) purchased from Sigma-Aldrich Chimie (Saint-Quentin-Fallavier, France). It was determined by the kinetic mode of the spectrophotometric determination method, which was performed by recording the rate of change in absorbance at 234 nm. Indeed, the increase in absorbance at 234 nm was due to the formation of 13-hydroperoxides of linoleic acid (substrate used for LOX inhibition activity assay) [41,42,43].

A stock solution of LOX was prepared by dissolving around 5.7 units/mL of LOX in PBS (Phosphate Buffer Solution; 1 unit corresponding to 1 µmol of hydroperoxide formed per min). Four concentrations of *I. dewevrei* leaf essential oil sample (S44) in dimethylsulfoxide (DMSO) were tested as the inhibitor solution for the LOX inhibition activity assay: 0.005, 0.010, 0.015 and 0.020 mg/mL.

The LOX inhibition assays were performed by mixing 10 µL of LOX solution with 10 µL of inhibitor solution in 970 µL of boric acid buffer (50 mM; pH 9.0) and incubating them briefly at room temperature. The reaction started by the addition of 10 µL of substrate solution (Linoleic acid, 25 mM), and the velocity was recorded for 30 s at 234 nm. One assay was measured in the absence of the inhibitor solution, and one assay was measured with DMSO mixed with distilled water (99.85% of DMSO in distilled water) which made it possible to eliminate the inhibition effect of DMSO. No inhibitor effect of DMSO on the LOX activity was detected, and the LOX activity measured without inhibitor solution was considered as a control (100% enzymatic reaction). All assays were performed in triplicate. The percentage of lipoxygenase inhibition was calculated according to the equation:Inhibition % = (V_0control_ − V_0sample_) × 100/V_0control_V_0control_ is the activity of LOX in the absence of the inhibitor solution, and V_0sample_ is the activity of LOX in the presence of the inhibitor solution [43]. The IC_50_ was calculated by the concentration of *I. dewevrei* leaf essential oil in DMSO inhibiting 50% of LOX activity.

## 4. Conclusions

The chemical variability of the leaf essential oil from *I. dewevrei* growing wild in Côte d’Ivoire was investigated through 47 oil samples. A combination of chromatographic (CC, GC(RI)) and spectroscopic (GC/MS, ^13^C-NMR) techniques was used to determine the samples’ chemical compositions. One hundred and thirteen constituents accounting for 90.8–98.9% of the whole sample compositions were identified, and the main components varied drastically from sample to sample. Therefore, the 47 oil compositions were submitted to hierarchical cluster and principal components analyses, which evidenced three distinct chemical groups, each dividing into two subgroups. The Subgroup I−A was dominated by (*Z*)-β-ocimene, β-eudesmol, germacrene D and (*E*)-β-ocimene, while (10βH)-1β,8β-oxido-cadina-4-ene, santalenone, *trans*-α-bergamotene and *trans*-β-bergamotene were the main compounds of Subgroup I−B. The prevalent constituents of Subgroup II−A were germacrene B, (*E*)-β-caryophyllene, (5αH,10βMe)-6,12-oxido-elema-1,3,6,11(12)-tetraene and γ-elemene. The Subgroup II−B displayed germacrene B, germacrene D and (*Z*)-β-ocimene as the majority compounds. Germacrene D was the most abundant constituent of Group III, followed in Subgroup III−A by (*E*)-β-caryophyllene, (10βH)-1β,8β-oxido-cadina-4-ene, germacrene D-8-one, and then in Subgroup III−B by (*Z*)-β-ocimene and (*E*)-β-ocimene. Compounds bearing the eudesmane skeleton characterized Subgroup I−A and also Subgroup III−B to a lesser extent. Likewise, santalane, bergamotane and bisabolane skeletons were markers of Subgroup I−B, while the elemane skeleton was specific to Group II. Group III markers were oxygenated germacrane and cadinane compounds. Although genetic factors could not be completely excluded, the observed qualitative and quantitative chemical variability of the leaf essential oil from *I. dewevrei* could be related to mostly phenology and season, then harvest site to a lesser extent. A leaf oil sample (S44) was tested for its lipoxygenase inhibition ability. The oil IC_50_ value (0.020 ± 0.005 mg/mL) was slightly higher than the non-competitive lipoxygenase inhibitor NDGA IC_50_ value (0.013 ± 0.003 mg/mL). Therefore, this leaf essential oil exhibited significant in vitro anti-inflammatory potential.

## Figures and Tables

**Figure 1 molecules-26-06228-f001:**
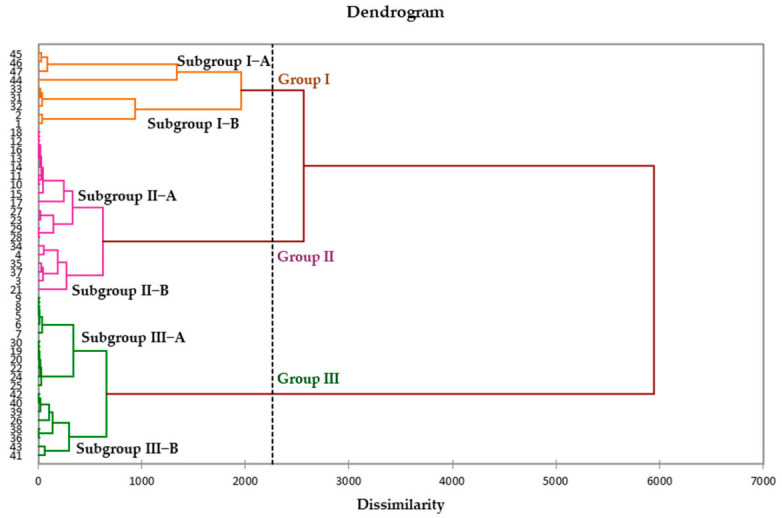
Dendrogram of hierarchical cluster analysis (HCA) of the 47 leaf oil samples from *I. dewevrei*.

**Figure 2 molecules-26-06228-f002:**
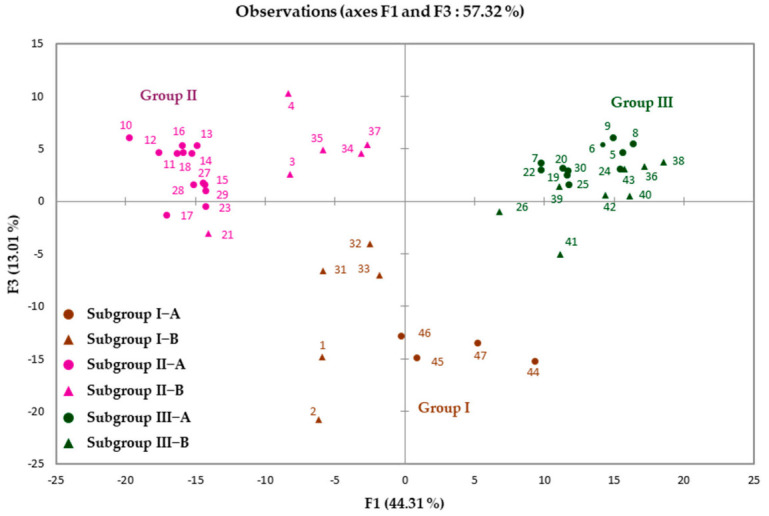
Principal component analysis (PCA) of the 47 leaf oil samples from *I. dewevrei*.

**Figure 3 molecules-26-06228-f003:**
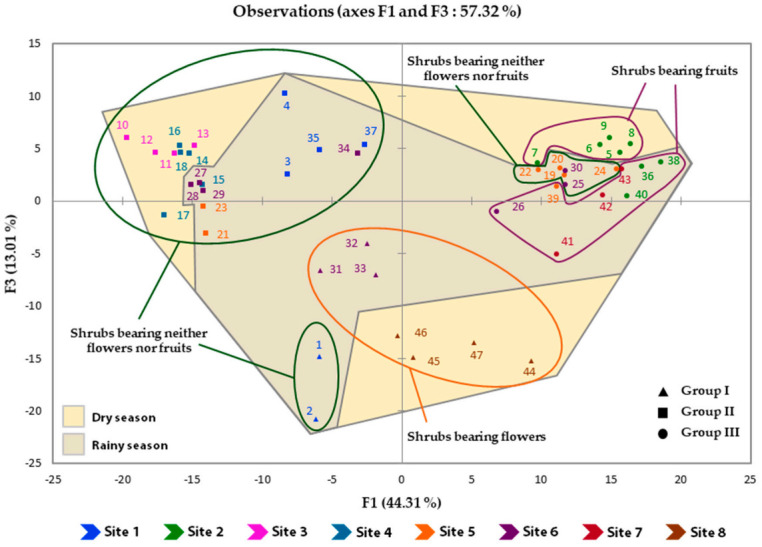
Effect of phenology, season and harvest site on the 47 oil samples’ distribution into groups.

**Table 1 molecules-26-06228-t001:** Chemical composition of five leaf essential oil samples from *Isolona dewevrei*.

N°	Compounds ^a^	RIl ^b^	RIa	RIp	RRF	S2 (%)	S14 (%)	S24 (%)	S44 (%)	S47 (%)	Identification
1	α-Thujene	932	923	1016	0.765	0.2	0.5	tr	0.1	0.2	RI, MS, ^13^C-NMR
2	α-Pinene	936	931	1013	0.765	0.1	0.4	0.1	tr	0.1	RI, MS, ^13^C-NMR
3	Sabinene	973	965	1120	0.765	0.6	1.8	0.1	0.2	1.0	RI, MS, ^13^C-NMR
4	β-Pinene	978	970	1109	0.765	tr	0.4	0.1	-	-	RI, MS, ^13^C-NMR
5	Myrcene	987	981	1158	0.765	0.2	0.6	0.3	0.9	0.3	RI, MS, ^13^C-NMR
6	α-Phellandrene	1002	997	1162	0.765	tr	0.1	tr	0.1	-	RI, MS
7	δ-3-Carene	1010	1005	1146	0.765	tr	0.1	tr	0.1	-	RI, MS
8	α-Terpinene	1013	1009	1178	0.765	tr	0.1	0.1	0.2	tr	RI, MS, *^13^C-NMR*
9	*p*-Cymene	1015	1012	1268	0.698	0.4	0.3	tr	0.1	-	RI, MS, ^13^C-NMR
10	Limonene	1025	1021	1199	0.765	0.1	2.4	1.1	0.2	tr	RI, MS, ^13^C-NMR
11	(*Z*)-β-Ocimene	1029	1025	1230	0.765	0.8	7.8	3.4	36.6	10.2	RI, MS, ^13^C-NMR
12	(*E*)-β-Ocimene	1041	1036	1247	0.765	0.5	4.8	4.5	30.5	3.4	RI, MS, ^13^C-NMR
13	γ-Terpinene	1051	1048	1242	0.765	0.2	0.3	0.2	0.5	0.1	RI, MS, ^13^C-NMR
14	Terpinolene	1082	1078	1279	0.765	tr	0.1	tr	0.1	-	RI, MS
15	Linalool	1086	1083	1543	0.869	0.1	0.2	tr	0.1	tr	RI, MS, *^13^C-NMR*
16	*allo*-Ocimene	1113	1117	1370	0.765	0.1	0.2	0.1	1.3	0.4	RI, MS, ^13^C-NMR
17	Terpinen-4-ol	1164	1161	1597	0.869	0.1	0.2	-	tr	0.1	RI, MS, *^13^C-NMR*
18	Neral	1215	1212	1679	0.887	-	0.1	-	tr	0.1	RI, MS
19	Geraniol	1235	1233	1843	0.869	0.2	-	0.1	tr	tr	RI, MS, *^13^C-NMR*
20	Geranial	1244	1244	1740	0.887	tr	tr	0.1	-	0.1	RI, MS
21	Thymol	1267	1267	2178	0.808	1.7	-	tr	0.2	-	RI, MS, ^13^C-NMR
22	Carvacrol	1278	1277	2219	0.808	-	tr	-	-	1.0	RI, MS, ^13^C-NMR
23	Eugenol	1331	1328	2170	0.808	-	-	-	-	0.4	RI, MS, ^13^C-NMR
24	Bicycloelemene	1338	1331	1485	0.751	-	0.2	-	-	0.1	RI, MS, *^13^C-NMR*
25	δ-Elemene	1340	1334	1464	0.751	-	3.9	tr	0.3	1.9	RI, MS, ^13^C-NMR
26	α-Cubebene	1355	1347	1452	0.751	-	0.1	0.1	0.2	-	RI, MS, ^13^C-NMR
27	α-Ylangene	1376	1368	1475	0.751	-	0.1	tr	0.1	0.1	RI, MS
28	α-Copaene	1379	1374	1485	0.751	-	0.2	0.9	0.8	0.1	RI, MS, ^13^C-NMR
29	β-Cubebene	1390	1384	1539	0.751	0.7	-	-	0.3	0.1	RI, MS, ^13^C-NMR
30	β-Elemene	1389	1385	1583	0.751	tr	3.1	1.6	1.3	1.2	RI, MS, ^13^C-NMR
31	α-Funebrene	1385	1386	1518	0.751	0.6	-	-	tr	-	RI, MS, ^13^C-NMR
32	Sesquithujene	1399	1400	1549	0.751	0.2	tr	-	tr	-	RI, MS, *^13^C-NMR*
33	Cyperene	1402	1404	1524	0.751	-	0.3	-	-	tr	RI, MS, *^13^C-NMR*
34	*cis*-α-Bergamotene	1411	1409	1561	0.751	2.3	tr	-	-	-	RI, MS, ^13^C-NMR
35	(*E*)-β-Caryophyllene *	1421	1416	1589	0.751	1.7	6.8	5.0	2.2	12.4	RI, MS, ^13^C-NMR
36	α-Santalene *	1422	1416	1565	0.751	8.5	tr	0.1	-	-	RI, MS, ^13^C-NMR
37	β-Copaene *	1430	1426	1574	0.751	0.3	-	-	0.2	0.1	RI, MS, *^13^C-NMR*
38	γ-Elemene *#	1429	1426	1630	0.751	-	6.9	tr	tr	0.3	RI, MS, ^13^C-NMR
39	*trans*-α-Bergamotene	1434	1431	1578	0.751	11.7	-	-	tr	tr	RI, MS, ^13^C-NMR
40	Sesquisabinene A	1435	1434	1636	0.751	1.3	-	-	-	tr	RI, MS, ^13^C-NMR
41	β-Sesquifenchene	1437	1439	1611	0.751	0.5	-	tr	-	-	RI, MS, ^13^C-NMR
42	*epi*-β-Santalene	1446	1441	1626	0.751	0.5	0.1	tr	-	-	RI, MS, ^13^C-NMR
43	(*E*)-β-Farnesene	1446	1446	1660	0.751	2.1	0.1	0.1	tr	-	RI, MS, ^13^C-NMR
44	α-Humulene	1455	1448	1662	0.751	0.8	1.2	1.3	0.5	1.9	RI, MS, ^13^C-NMR
45	β-Santalene	1460	1453	1643	0.751	0.2	-	tr	-	-	RI, MS, *^13^C-NMR*
46	(5αH,10βMe)-6,12-Oxido- elema-1,3,6,11(12)-tetraene #	1455 ^c^	1455	1837	0.853	-	10.3	0.1	-	1.8	RI, MS, ^13^C-NMR
47	Ishwarane	1468	1460	1644	0.751	0.2	-	-	-	-	RI, MS, *^13^C-NMR*
48	β-Acoradiene	1465	1461	1669	0.751	0.2	-	-	tr	-	RI, MS, *^13^C-NMR*
49	6,12-Oxido-germacra- 1(10),4,6,11(12)-tetraene #	1463 ^c^	1463	1845	0.853	-	1.0	-	-	0.2	RI, MS, ^13^C-NMR
50	α-Curcumene	1473	1469	1766	0.707	1.1	0.2	tr	tr	-	RI, MS, ^13^C-NMR
51	γ-Muurolene	1474	1471	1683	0.751	0.8	2.5	0.3	0.3	0.2	RI, MS, ^13^C-NMR
52	Germacrene D	1479	1474	1700	0.751	0.2	7.5	23.6	13.8	16.3	RI, MS, ^13^C-NMR
53	*trans*-β-Bergamotene	1480	1478	1676	0.751	10.9	0.5	tr	-	-	RI, MS, ^13^C-NMR
54	β-Selinene	1486	1484	1710	0.751	0.1	0.1	0.1	tr	1.8	RI, MS, ^13^C-NMR
55	α-Zingiberene	1489	1485	1712	0.751	0.4	-	-	0.1	-	RI, MS, ^13^C-NMR
56	α-Selinene	1494	1490	1723	0.751	0.1	-	-	-	1.4	RI, MS, ^13^C-NMR
57	Bicyclogermacrene *	1494	1491	1721	0.751	0.1	0.1	1.8	0.7	-	RI, MS, ^13^C-NMR
58	α-Muurolene *	1496	1491	1720	0.751	tr	-	-	0.2	0.1	RI, MS, *^13^C-NMR*
59	(*Z*)-α-Bisabolene	1494	1492	1724	0.751	0.1	0.1	-	-	-	RI, MS
60	γ-Cadinene	1507	1493	1753	0.751	-	0.1	0.2	0.1	0.1	RI, MS, *^13^C-NMR*
61	β-Bisabolene	1503	1500	1719	0.751	8.7	0.1	0.2	tr	-	RI, MS, ^13^C-NMR
62	(*E,E*)-α-Farnesene	1498	1501	1748	0.751	0.7	0.3	-	-	-	RI, MS, ^13^C-NMR
63	β-Curcumene	1503	1502	1736	0.751	0.4	-	-	-	-	RI, MS, ^13^C-NMR
64	(*Z*)-γ-Curcumene	1493	1506	1732	0.751	0.8	0.2	-	-	-	RI, MS, ^13^C-NMR
65	β-Sesquiphellandrene	1516	1512	1766	0.751	0.4	-	-	-	-	RI, MS, ^13^C-NMR
66	δ-Cadinene	1520	1514	1753	0.751	-	tr	2.5	0.8	0.3	RI, MS, ^13^C-NMR
67	*cis*-Lanceol	1525 ^d^	1517	2087	0.819	0.3	tr	0.9	0.2	-	RI, MS, ^13^C-NMR
68	(*Z*)-γ-Bisabolene	1505	1521	1721	0.751	0.9	0.6	1.4	0.2	-	RI, MS, ^13^C-NMR
69	*trans*-Sesquisabinene hydrate	1530 ^e^	1530	1984	0.819	2.0	0.3	tr	0.1	-	RI, MS, ^13^C-NMR
70	(*E*)-α-Bisabolene	1530	1531	1761	0.751	0.6	tr	-	-	-	RI, MS, ^13^C-NMR
71	(10βH)-1β,8β-Oxido-cadin-4-ene	1534 ^f^	1534	1853	0.830	9.3	0.4	7.3	1.3	-	RI, MS, ^13^C-NMR
72	β-Elemol	1541	1536	2077	0.819	0.4	0.7	tr	1.2	1.7	RI, MS, ^13^C-NMR
73	(*E*)-Nerolidol	1553	1547	2034	0.819	1.1	0.1	0.5	0.3	-	RI, MS, ^13^C-NMR
74	Germacrene B #	1552	1549	1818	0.751	-	20.8	0.4	0.1	2.5	RI, MS, ^13^C-NMR
75	Santalenone	1576	1560	1980	0.841	13.0	tr	-	-	-	RI, MS, ^13^C-NMR
76	*cis*-Sesquisabinene hydrate	1565 ^e^	1562	2079	0.819	0.8	tr	0.3	0.1	-	RI, MS, ^13^C-NMR
77	Germacra-1(10),5-dien-4β-ol	1572 ^g^	1564	2047	0.819	0.1	tr	tr	0.1	-	RI, MS, *^13^C-NMR*
78	Caryophyllene oxide	1578	1567	1973	0.830	0.3	0.8	0.1	-	0.1	RI, MS, ^13^C-NMR
79	Guaiol	1593	1581	2119	0.819	2.3	tr	tr	-	tr	RI, MS, ^13^C-NMR
80	Germacrene D-8-one	1588 ^h^	1584	2066	0.841	0.5	0.1	8.7	0.6	tr	RI, MS, ^13^C-NMR
81	Humulene oxide II	1602	1597	2042	0.830	0.1	tr	0.4	-	-	RI, MS, ^13^C-NMR
82	*epi*-Cubenol	1602	1605	2046	0.819	0.2	0.3	-	-	-	RI, MS, *^13^C-NMR*
83	Alismol	1619	1609	2245	0.830	-	0.6	0.1	-	-	RI, MS, *^13^C-NMR*
84	Eremoligenol	1614	1614	2196	0.819	0.3	-	-	-	0.5	RI, MS, ^13^C-NMR
85	γ-Eudesmol	1618	1620	2172	0.819	tr	0.9	1.2	tr	0.4	RI, MS, ^13^C-NMR
86	Muurola-4,10(14)-dien-8β-ol	1632 ^i^	1629	2186	0.830	0.9	0.1	3.2	0.3	-	RI, MS, ^13^C-NMR
87	β-Eudesmol	1641	1635	2225	0.819	tr	0.2	-	-	22.5	RI, MS, ^13^C-NMR
88	α-Cadinol	1643	1637	2231	0.819	0.2	0.7	0.6	tr	-	RI, MS, ^13^C-NMR
89	α-Eudesmol	1653	1639	2216	0.819	-	0.1	-	-	12.4	RI, MS, ^13^C-NMR
90	Tumerone-ar	1643	1642	2252	0.841	-	-	0.1	-	0.1	RI, MS
91	β-Bisabolol	1659	1653	2144	0.819	1.1	0.1	0.2	tr	-	RI, MS, ^13^C-NMR
92	(7αH)-Germacrene D-8α-ol *	1657 ^f^	1657	2355	0.819	tr	-	7.8	0.7	-	RI, MS, ^13^C-NMR
93	(7αH)-Germacrene D-8β-ol *	1657 ^f^	1657	2355	0.819	tr	-	2.6	0.2	-	RI, MS, ^13^C-NMR
94	α-Bisabolol	1673	1664	2208	0.819	0.7	tr	1.4	0.1	-	RI, MS, ^13^C-NMR
95	*epi*-α-Bisabolol	1667 ^j^	1667	2214	0.819	0.7	tr	0.1	-	-	RI, MS, ^13^C-NMR
96	Cadina-1(10),4-dien-8β-ol	1676 ^f^	1676	2276	0.819	tr	0.3	7.6	0.3	-	RI, MS, ^13^C-NMR
97	Cadina-4,10(14)-dien-8β-ol	1675 ^i^	1678	2280	0.830	0.1	0.4	0.8	-	-	RI, MS, ^13^C-NMR
98	(1βH,5βH)-6,12-Oxido-guaia- 6,10(14),11(12)-trien-4α-ol	1754 ^c^	1754	2519	0.853	tr	1.7	-	-	0.1	RI, MS, ^13^C-NMR
99	(6αH)-Germacra-1(10),4,7(11)- trien-6,12-γ-lactone	1856 ^c^	1856	2829	0.985	-	0.8	-	-	-	RI, MS, ^13^C-NMR
	Hydrocarbon monoterpenes					3.2	19.9	10.0	70.9	15.7	
	Oxygenated monoterpenes					2.1	0.5	0.2	0.3	1.3	
	Hydrocarbon sesquiterpenes					58.1	56.1	39.6	22.2	40.9	
	Oxygenated sesquiterpenes					34.4	19.9	44.0	5.5	39.8	
	Other compounds					-	-	-	-	0.4	
	Total					97.8	96.4	93.8	98.9	98.1	

^a^ Order of elution and percentages are given on an apolar column (BP-1), except components with an asterisk (*), where percentages are taken on a polar column (BP-20). (#) Thermolabile compound, percentage evaluated by a combination of GC-FID and ^13^C-NMR data [23,24]. ^b^ RIl: Retention indices reported in the Terpenoids Library Website [25] or in reference ^c^ [15]; ^d^ [26]; ^e^ [27]; ^f^ [16]; ^g^ [28]; ^h^ [14]; ^i^ [29]; ^j^ [30]. RIa, RIp: retention indices measured on apolar and polar capillary column, respectively. RRF: relative response factors calculated using methyl octanoate as internal standard. The relative proportions of constituent are expressed in g/100 g. (-): not detected; tr: traces level (<0.05%). S2: sample 2; the same for S14, S24, S44 and S47. ^13^C-NMR: compounds identified by NMR in the essential oil samples and obvious in at least one fraction of chromatography; *^13^C-NMR (italic)*: compounds identified by NMR in fractions of chromatography.

**Table 2 molecules-26-06228-t002:** Chemical variability of the leaf essential oil from *Isolona dewevrei* (main constituents).

Component ^[a]^	RIa ^[b]^	RIp ^[b]^	Group I						Group II						Group III					
			Subgroup I−A			Subgroup I−B			Subgroup II−A			Subgroup II−B			Subgroup III−A			Subgroup III−B		
			M% ± SD	Min	Max	M% ± SD	Min	Max	M% ± SD	Min	Max	M% ± SD	Min	Max	M% ± SD	Min	Max	M% ± SD	Min	Max
Limonene	1021	1199	0.1 ± 0.1	tr	0.2	0.8 ± 0.6	0.1	1.8	**4.4 ± 3.4**	**1.8**	**13.0**	1.0 ± 1.0	0.1	2.1	2.5 ± 1.8	1.1	6.3	0.1 ± 0.2	tr	0.4
(*Z*)-β-Ocimene	1025	1230	**18.0 ± 12.5**	**10.2**	**36.6**	0.7 ± 0.4	0.2	1.1	**5.9 ± 2.7**	**0.1**	**9.2**	**13.2 ± 5.2**	**7.8**	**22.7**	**4.5 ± 1.6**	**3.0**	**8.8**	**12.1 ± 2.0**	**9.4**	**14.0**
(*E*)-β-Ocimene	1036	1247	**11.4 ± 12.8**	**3.4**	**30.5**	0.6 ± 0.4	0.2	1.2	3.7 ± 1.4	-	5.1	**4.6 ± 1.6**	**1.8**	**6.2**	3.8 ± 1.3	2.2	6.4	**7.0 ± 1.6**	**4.9**	**9.1**
(*E*)-β-Caryophyllene *	1416	1589	**8.0 ± 2.5**	**2.2**	**12.4**	3.0 ± 0.9	1.7	3.8	**9.0 ± 1.8**	**6.8**	**12.1**	**7.5 ± 2.5**	**3.4**	**10.7**	**8.2 ± 2.8**	**5.0**	**11.6**	**5.2 ± 2.1**	**2.6**	**8.1**
α-Santalene *	1416	1565	-	-	-	**5.2 ± 1.8**	**4.1**	**8.5**	tr	-	0.1	tr	-	0.1	0.1 ± 0.1	-	0.2	tr	-	0.1
γ-Elemene #	1426	1630	0.3 ± 0.2	-	0.5	0.8 ± 0.6	tr	1.3	**6.3 ± 1.3**	**4.0**	**7.9**	**4.2 ± 2.4**	**1.4**	**7.5**	0.4 ± 0.4	tr	1.1	0.4 ± 0.1	0.1	0.5
*trans*-α-Bergamotene	1431	1578	-	-	-	**7.8 ± 2.5**	**6.0**	**11.7**	tr	-	tr	tr	-	0.1	tr	-	tr	tr	-	tr
(5αH,10βMe)-6,12-Oxido-elema-1,3,6,11(12)-tetraene #	1455	1837	1.3 ± 0.9	-	1.8	0.6 ± 0.6	-	1.4	**7.8 ± 3.1**	**3.2**	**13.7**	**6.3 ± 4.4**	**tr**	**10.6**	0.9 ± 0.8	-	2.2	1.1 ± 0.7	0.4	2.2
Germacrene D	1474	1700	**12.6 ± 3.0**	**9.9**	**16.3**	1.7 ± 1.1	0.2	2.8	**5.7 ± 2.0**	**0.9**	**8.1**	**13.6 ± 5.2**	**4.0**	**17.8**	**24.1 ± 3.4**	**20.5**	**29.2**	**27.6 ± 4.1**	**23.1**	**34.0**
*trans*-β-Bergamotene	1478	1676	-	-	-	**7.8 ± 2.0**	**6.1**	**10.9**	0.6 ± 0.2	0.4	1.1	0.2 ± 0.2	-	0.5	0.1 ± 0.0	tr	0.1	tr	-	0.2
β-bisabolene	1500	1719	-	-	-	**5.0 ± 2.6**	**2.6**	**8.7**	0.1 ± 0.1	0.1	0.2	tr	-	0.1	0.3 ± 0.1	0.2	0.4	0.1 ± 0.2	-	0.5
(10βH)-1β,8β-Oxido-cadina-4-ene	1534	1853	0.3 ± 0.7	-	1.3	**10.9 ± 2.1**	**8.7**	**13.7**	0.4 ± 0.2	0.2	0.9	1.3 ± 1.2	-	2.7	**6.4 ± 0.5**	**5.5**	**7.3**	**4.0 ± 2.8**	**-**	**6.2**
Germacrene B #	1549	1818	2.6 ± 1.8	0.1	4.1	1.7 ± 1.1	-	2.8	**18.5 ± 4.1**	**12.2**	**24.5**	**17.2 ± 4.1**	**13.2**	**24.8**	2.0 ± 1.1	0.4	4.0	2.3 ± 1.6	0.6	4.5
Santalenone	1560	1980	-	-	-	**9.0 ± 2.7**	**6.9**	**13.0**	tr	-	0.1	tr	-	0.1	-	-	-	-	-	-
Germacrene D-8-one	1584	2066	0.2 ± 0.3	tr	0.6	0.7 ± 0.3	0.3	1.1	0.1 ± 0.1	-	0.3	0.6 ± 0.6	tr	1.2	**6.1 ± 2.2**	**3.2**	**8.7**	2.3 ± 1.6	tr	3.6
β-Eudesmol	1635	2225	**13.4 ± 9.7**	**-**	**22.5**	0.3 ± 0.3	-	0.8	0.0 ± 0.1	-	0.2	tr	-	tr	-	-	-	2.4 ± 3.7	-	9.2
α-Eudesmol	1639	2216	**7.6 ± 5.5**	**-**	**12.4**	0.4 ± 0.4	-	0.9	tr	-	0.1	tr	-	tr	-	-	-	1.5 ± 2.2	-	5.6
(7βH)-Germacrene D-8α-ol #	1657	2355	0.2 ± 0.4	-	0.7	0.2 ± 0.2	tr	0.5	0.1 ± 0.1	-	0.3	0.9 ± 0.6	tr	1.5	**4.7 ± 2.3**	**2.0**	**7.8**	2.0 ± 1.5	-	3.6
Cadina-1(10),4-dien-8β-ol	1676	2276	0.1 ± 0.2	-	0.3	0.9 ± 0.8	tr	1.9	0.3 ± 0.2	0.2	0.7	1.4 ± 1.2	0.1	3.2	**5.7 ± 1.2**	**4.1**	**7.6**	3.5 ± 2.6	-	6.2
Essential oil extraction Yields (%)			0.3 ± 0.2	0.1	0.5	0.3 ± 0.1	0.2	0.4	0.1 ± 0.0	0.1	0.2	0.3 ± 0.1	0.1	0.4	0.2 ± 0.1	0.2	0.3	0.3 ± 0.1	0.1	0.4

^[a]^ Order of elution and percentages on apolar column (BP-1), except components with an asterisk (*), percentages on polar column (BP-20); (#) percentages calculated by combination of GC(FID) and ^13^C NMR; ^[b]^ RIa, RIp: Retention indices measured on apolar and polar capillary column, respectively; M% ± SD: mean percentage and standard deviation; (-): not detected; tr: traces level (<0.05%).

**Table 3 molecules-26-06228-t003:** IC_50_ values and in vitro anti-inflammatory activity of *Isolona dewevrei* leaf essential oil.

Percentage Inhibition of LOX				IC_50_ (mg/mL)	
Concentration #	Inhibition (%)	Concentration #	Inhibition (%)	Essential oil	0.020 ± 0.005
0.005	10.3 ± 0.8	0.015	47.9 ± 10.4	* NDGA	0.013 ± 0.003
0.010	12.7 ± 0.7	0.020	51.5 ± 13.8		

Values are means of triplicates ± standard deviation; * NDGA: NorDihydroGuaiaretic Acid; # mg/mL.

**Table 4 molecules-26-06228-t004:** Weights of fractions from column chromatography.

Samples	Fractions Weights (g)							Total (g)
	F1 P: 100%	F2 P: 100%	F3 P/DE: 98/2%	F4 P/DE: 95/5%	F5 P/DE: 90/10%	F6 P/DE: 80/20%	F7 DE: 100%	
S2 (4.115 g)	1.301	1.187	0.197	0.402	0.489	0.212	0.307	4.095
S14 (4.820 g)	2.470	1.109	0.144	0.285	0.413	0.141	0.190	4.752
S24 (3.702 g)	1.221	0.506	0.201	0.478	0.615	0.246	0.352	3.619
S44 (4.226 g)	2.543	1.371	0.031	0.089	0.082	0.029	0.032	4.177
S47 (3.910 g)	1.256	0.940	0.184	0.471	0.521	0.179	0.312	3.863

P: *n*-pentane; DE: Diethyl ether.

## Data Availability

The data presented in this study are available in Appendix A.

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
