# Peer review of "Chemical Variability and In Vitro Anti-Inflammatory Activity of Leaf Essential Oil from Ivorian Isolona dewevrei (De Wild. & T. Durand) Engl. & Diels"

_molecules, 2021, doi:10.3390/molecules26206228_

Round 1

Reviewer 1 Report

The article is based on the obtaining of natural oils from Isolona dewevrei, and their subsequent identification and anti-inflammatory evaluation.
1. Include images of the study plant
2. Indicate how much plant material was used to obtain the essential oils

Author Response

Dear Editor,

We provided a revised version of the manuscript. We considered all the remarks of referee and we revised the manuscrit highlighting the corrections.

Editor

  1. In addition to reviewers comments I would suggest to add the RI from litterature in Table 1.

The RI from litterature were added in Table 1.

  1. The impact factors on the observed qualitative and quantitative chemical variability must be added in the summary (phenology , others?).

We already added in the abstract and the conclusion, the impact of phenology, season and harvest site on the observed qualitative and quantitative chemical variability.

  1. Line 380 : could author be more precise (temperature, rainfall, humidity of the microclimates ? ) in order to try and answer the question?

Climate data (Temperature, rainfall and humidity) were added in Table S2 of the Supplementary material.

Reviewer 1

  1. Include images of the study plant

Image of the study plant was already included in the graphical abstract.

  1. Indicate how much plant material was used to obtain the essential oils

Plant material and essential oil extraction data were already provided in the Table S2 of the Supplementary material.

Reviewer 2

  1. Title: you should include in vitro antiinflammatory activity

The title was modified as suggested.

  1. Abstract: you explain a lot about the chemical studies but there is poor information about biological activity.

The abstract was improved.

  1. Results and discussion: you need to explain more why you chose the selected samples, review some tables and other comments you will find at the document added.

All the recommandations were taken into account.

  1. Material and methods: you will find some comments along the document but pay special attention to anti-inflammatory assay. You need to clarify what sample you are showing the results about.

Modifications were made according to the suggestions.

  1. Conclusions: please add specific info about anti-inflammatory activity. 

The conclusion was improved.

  1. References: please check reference number 16.

Reference number 16 was modified.

We hope that our revised manuscript is now suitable for publication in this special issue of Molecules.

Best regards,

Prof. Félix Tomi

Reviewer 2 Report

Dear Authors:

Congrats for your well and hard work, I know this kind of activity.

You will see all my comments and suggestions at the attached file, however I resume as:

  • title: you should include in vitro antiinflammatory activity
  • abstract: you explain a lot about the chemical studies but there is poor information about biological activity.
  • results and discussion: you need to explain more why you chose the selected samples, review some tables and other comments you will find at the document  added.
  • material and methods: you will find some comments along the document but pay special attention to anti-inflammatory assay. You need to clarify what sample you are showing the results about.
  • conclusions: please add specific info about anti-inflammatory activity. 
  • references: please check reference number 16

Please check all manuscript to harmonize the info added (for example in vitro, etc.)

Good luck, you may only make a little effort to see your paper published.

All my best,

Author Response

(The authors gave the same response as above.)
